# Distribution of common pipistrelle (*Pipistrellus pipistrellus*) activity is altered by airflow disruption generated by wind turbines

**Camille Leroux** [1,2]*, **Kévin Barré** [1], **Nicolas Valet** [2], **Christian Kerbiriou** [1]☯, **Isabelle Le Viol** [1]☯

**1** Centre d'Ecologie et des Sciences de la Conservation (CESCO), Muséum National d'Histoire Naturelle, Centre National de la Recherche Scientifique, Sorbonne Université, Station Marine, Concarneau, France, **2** Auddicé biodiversité– ZAC du Chevalement, Roost-Warendin, France

☯ These authors contributed equally to this work.
* Camille.leroux@edu.mnhn.fr, camille.leroux@auddice.com

**Data Availability Statement:** Data used in the analysis are available at https://doi.org/10.5281/zenodo.8358475.

## Abstract

The mechanisms underlying bat and bird activity peaks (attraction) or losses (avoidance) near wind turbines remain unknown. Yet, understanding them would be a major lever to limit the resulting habitat loss and fatalities. Given that bat activity is strongly related to airflows, we hypothesized that airflow disturbances generated leeward (downwind) of operating wind turbines–via the so-called wake effect–make this area less favorable for bats, due to increased flight costs, decreased maneuverability and possibly lower prey abundance. To test this hypothesis, we quantified *Pipistrellus pipistrellus* activity acoustically at 361 site-nights in western France in June on a longitudinal distance gradient from the wind turbine and on a circular azimuth gradient of wind incidence angle, calculated from the prevailing wind direction of the night. We show that *P. pipistrellus* avoid the wake area, as less activity was detected leeward of turbines than windward (upwind) at relatively moderate and high wind speeds. Furthermore, we found that *P. pipistrellus* response to wind turbine (attraction and avoidance) depended on the angle from the wake area. These findings are consistent with the hypothesis that changes in airflows around operating wind turbines can strongly impact the way bats use habitats up to at least 1500 m from the turbines, and thus should prompt the consideration of prevailing winds in wind energy planning. Based on the evidence we present here, we strongly recommend avoiding configurations involving the installation of a turbine between the origin of prevailing winds and important habitats for bats, such as hedgerows, water or woodlands.

## Introduction

Wind energy is a challenging dilemma: while its use may mitigate climate change, wind turbines have adverse impacts on biodiversity. In flying vertebrates, wind turbines have been documented to alter migration and commuting routes, the choice of stopover sites, and the use of surrounding habitats [1–3]. Specifically, wind turbines can attract or be avoided by birds [4] and bats [5]. Attraction–defined here as an increase in activity near wind turbines—can

**Funding:** The Association Nationale de la Recherche et de la Technologie (Grant No. 2019/1566) and Auddicé biodiversité funded this research. Wind farm developers funded part of the bat recorders. Kévin Barré was funded by the Agence de la transition écologique (ADEME), Christian Kerbiriou by Sorbonne University, and Isabelle Le Viol by the french National Museum of Natural History (MNHN). Auddicé biodiversité was involved in both direct funding and funding acquisition, projet administration, supervision and preparation of the manuscript. Other funders had no role in study design, data collection and analysis, decision to publish, or preparation of the manuscript.

**Competing interests:** This work is part of the first author's PhD research, which was co-supervised by all the co-authors from the National Museum of Natural History (MNHN) and Auddicé biodiversité. Auddicé biodiversité is an environmental consultancy that conducts wind farm impact assessment studies. At the time of submission two of the authors - Camille Leroux and Nicolas Valet - were working at Auddicé biodiversité. In addition, Kévin Barré was funded by ADEME, a public agency promoting renewable energies. Members of the wind energy sector financed part of the bat recorders and provided some technical data and expertise on wind turbine operation and features, as stated above. Thus, the authors declare a potential conflict of interest. However, sampling design, acoustic data collection, analysis and writing were conducted only by the authors, and members of the wind energy sector did not view the draft before submission. Furthermore, sampling design and sampling sites were determined independently from Auddicé biodiversité activities and identification of bat echolocation calls and bat activity measures were provided by TADARIDA software, a MNHN web portal. The authors certify that the collaboration did not interfere with the stated hypothesis, the way it was tested or the interpretations and conclusions. Authors take full responsibility for the integrity of the study.

increase fatalities due to collisions, and avoidance–defined here as a decrease of activity near wind turbines–can cause loss of habitat use [6–8]. Some studies suggest that responses may vary depending on the sex of the individual [9], interindividual variation [10], wind turbine size [11], season [11], and habitat [5]. However, the mechanisms triggering attraction or avoidance are still elusive.

We suggest that abiotic factors such as airflows can play a primary role as well. Indeed, operating wind turbines generate massive airflow disturbances–called wake effect–by increasing turbulence and decreasing wind speed over hundreds of meters to leeward (downwind) [12]. Furthermore, wind speed immediately windward (upwind) of a wind turbine is reduced due to a blockage effect [12]. Research on attraction to and avoidance of wind turbines by bats has not accounted directly for the wake effect so far (e.g. by modeling it), and only one study has accounted for it indirectly (e.g. by considering proxies such as combined blade speed rotation, wind speed and wind direction) [13]. However, this study was not focused on the wake effect and thus the conducted analysis did not allow to assess in details the effects of the wake on bat activity distribution depending on different wind speed.

Bats can sense airflows and orient themselves by using them, thanks to aerodynamic feedback from tactile receptors associated with their wing hairs [14]. Previous study has shown that bats approach turbine nacelle through its leeward side, suggesting that they seek for calm areas at a local scale around turbines [15]. We hypothesized that bat distribution may also be shaped by airflow disturbances around wind turbines at a larger scale (i.e., at a landscape scale, in contrast with a local scale focusing on bat activity at the turbine only). To test this hypothesis, we assessed spatial variation in *Pipistrellus pipistrellus* activity around wind turbines through acoustic surveys at hedgerows (Fig 1A)–major commuting and foraging habitats for bats [16,17] –at 361 site-nights in western France in June. Since the wake effect and associated turbulence are very complex to model in 3D and no study has so far assessed its consequences on bats, we propose in this study a first assessment of its effect at ground height and based on basic spatial variables. Specifically, each sampling site was placed on a gradient of distance from wind turbines (i.e., a longitudinal gradient) and on a gradient of location around the turbine in relation to wind direction (i.e., on a circular azimuth gradient describing wind incidence angle), and sampling took place in different wind conditions (Fig 1B and 1C). In undisturbed airflow conditions (no wind or very low wind speed, stationary or low spinning blades), we expected bat activity levels to be homogeneous around the turbine at all wind incidence angles. In contrast, in windy conditions, spinning blades generate large-scale airflow turbulence leeward of the turbine, resulting in conditions that probably require more energy for flight than in the windward area. Thus, we expected bats to seek calmer areas with no or low turbulence anywhere windward of or lateral to the turbine or immediately leeward of the nacelle, where they would also benefit from reduced wind speed. Finally, regardless of wind conditions, we expected a heterogeneous spatial distribution of bat activity along the distance gradient: either more activity near the turbine (interpreted as attraction) or less activity near the turbine (interpreted as avoidance; [5,6]).

## Materials and methods

### Study area and sampling design

We sampled bat activity at 67 wind turbines on 20 wind farms, 1 to 6 wind turbines at each wind farm (mean ± standard deviation: 3 ± 3, n = 20), in western France (S1 Fig in S1 File) from 27 May to 30 June 2020, during the reproduction period. Sampled wind farms included from 4 to 11 turbines. The mean hub height of the 67 wind turbines was 95 m (± 12 m); the mean rotor diameter was 86 m (± 12 m). To test the hypothesis that bat activity is related to

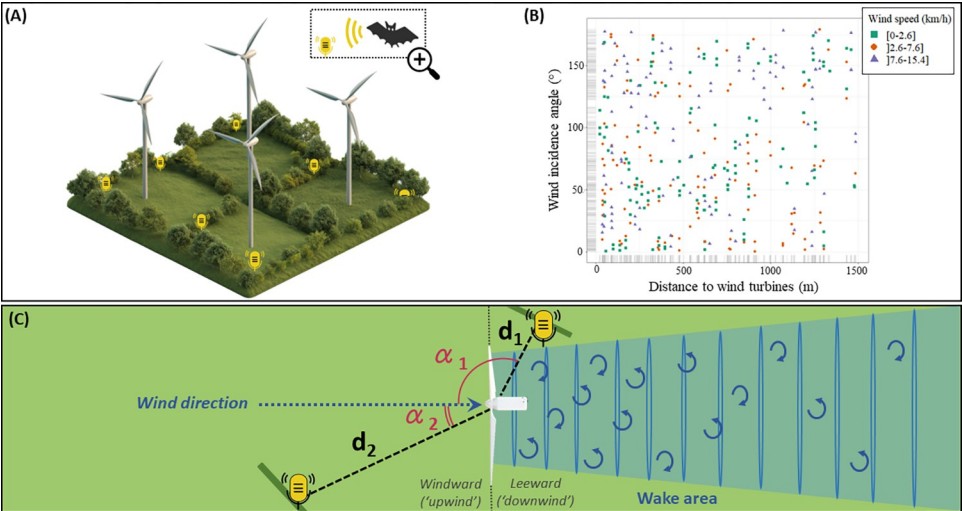

**Fig 1. Overview of the sampling design.** Schematic representation (not to scale) of acoustic sampling sites (shown as microphones) used to record bat activity each sampling night in hedgerows around a wind farm (A); distribution of the 361 sampled site-nights in relation to gradients of distance from the turbine, wind incidence angle and mean wind speed (B); and schematic top view representation (not to scale) of the wind incidence angle ($\alpha$) and the distance from the wind turbine (**d**) of two sampling sites (C).

airflow disturbances around turbines, we conducted acoustic sampling at 154 sites at hedgerows–defined here as tall linear landscape features including shrubs and trees–around the 67 turbines; each site was placed on a longitudinal gradient of distance from the wind turbine (23 to 1485 m; S2 Fig in S1 File) and on a circular azimuth gradient of wind incidence angle (S3 Fig in S1 File). The wind incidence angle was defined as the angle between the prevailing wind direction on the night of sampling and the line between the sampling site and the nearest wind turbine (from 0° for a site located windward–upwind–of the turbine, to 180° for a site located leeward–downwind; Fig 1C). The gradient of distance from the wind turbine (from 23 to 1484 m) was chosen in line with previous studies (effect of this distance detected up to at least 1000 m, [8]). It covers an area approximately seven times the average home range size of the species [18]. Each site was sampled on between one and four consecutive nights (2.3 ± 1.0, n = 154) to obtain inter-night variations in blade rotation speed (S4 Fig in S1 File), wind speed, and wind direction at each site, resulting in 361 site-nights sampled from 27 May to 30 June 2020. Sites were always located outside the convex polygon of a wind farm. Thus, the nearest wind turbine to the sampled site was located in the external perimeter of the wind farm. Each night, we sampled a complete gradient of distance to wind turbines around one or two (1.6 ± 0.5) wind farms. Our sampling design allowed us to obtain data for a complete circular azimuth gradient of wind incidence angle for each dataset, as well as good nesting of this gradient within the longitudinal gradient of distance from the wind turbine (Fig 1B and S5 Fig in S1 File).

## Acoustic data collection and analysis

At each site we used one SM4BAT-FS bat detector (Wildlife Acoustics Inc., Concord, MA, USA) coupled with one SMM-U2 ultrasonic microphone to record bat activity from 30 min before sunset until 30 min after sunrise. The microphones were placed 1.70 m above the ground. Sounds were automatically detected, recorded, and assigned to a bat species using TADARIDA software [19], which is widely used in french studies and sometimes in european studies [20–22]. We choose to focus on *Pipistrellus pipistrellus*, the most abundant species in

our dataset and a species whose behavior is widely known to be affected by wind turbines [6,23,24] and which features frequently in European reports of wind energy fatalities [25]. *P. pipistrellus* is an insectivorous bat feeding mainly on insects from Diptera and Lepidoptera orders [26] and foraging mainly near edges in the "aerial mode" [27]. To account for possible errors introduced by automated identification and to ensure the robustness of our results, we included in the analysis only sounds with a tolerance threshold of error risk $\leq 0.5$, and we checked that the results did not change for a more conservative threshold of $\leq 0.1$ (i.e., a reduction of false positives at the cost of discarding true positives; [28]). Regardless of the threshold, false positive and false negative rates are extremely low for *P. pipistrellus* in this area ($<1\%$ and $<0.1\%$, respectively; [28]), giving us confidence in the absence of bias associated with automated identification. We defined a bat pass as one or more echolocation calls attributed to *P. pipistrellus* within a 5-s interval, and used the total number of bat passes per night to quantify bat activity. Some bat activity was detected on every night. Based on the literature, we assumed that acoustic detection range did not differ significantly between both sides of the sampled hedgerow (see SI Appendix, Acoustic detection range).

## Statistical analysis

As our hypotheses were based on airflow disturbances that strongly depend on the wind speed, we split our dataset and we assessed bat spatial distribution for three balanced subsets (quantile classification) based on the mean wind conditions at a ground level (data extracted from meteociel) of the sampled night: (i) no wind or relatively low mean wind speed ([0–2.6] km/h of average wind speed and [1.7–9.4] km/h of average wind gusts; 126 site-nights), (ii) relatively moderate mean wind speed (]2.6–7.6] km/h and [5.8–18.6] km/h of average wind gusts; 120 site-nights) and (iii) relatively high mean wind speed (]7.6–15.4] km/h and [15.9–30.4] km/h of average wind gusts; 112 site-nights) (S6-S8 Figs in S1 File). We removed three outliers from the dataset for which the blade speed rotation was largely inferior to the rest of the values (36, 37 and 38 km/h) regarding the other data for a same mean wind speed–likely due to the maintenance or curtailment of this turbine on that night. The mean blade rotation speed for each of these datasets was 144 ($\pm$ 65), 161 ($\pm$ 61) and 188 ($\pm$ 24) km/h, respectively.

We modelled bat activity for each of these three datasets by using generalized linear mixed models (GLMMs; R package glmmTMB). We used a negative binomial error distribution (*nbinom1)*. In each model we performed, we checked that there was no collinearity issues using the *check_collinearity* function (r package performance), no overdispersion (*testDispersion* function, r package DHARMa) and we also checked model assumptions by plotting the residuals using the package DHARMa (*simulateResiduals* function). We first built a full model in which we included the sampling site as a random effect to control for repeated measures. We included as fixed effects the distance from the wind turbine, the wind incidence angle and their interaction. To account for wind direction variation and thus wind incidence angle variation within a night, we gave more weight to site-nights for which the prevailing wind direction was most closely represented during the night (see SI Appendix, Statistical Analysis, *Model weight calculation*). We also included as fixed effects the hedgerow length in a 1500 m buffer zone, since hedgerows may affect airflows; and the wind turbine rotor diameter, mean wind speed, and blade speed (see SI Appendix, Statistical Analysis, *Covariable extraction*), as they strongly drive both the spatial extent of the wake and the intensity of the turbulence that is generated (in our datasets, wind speed and blade rotation speed were not correlated, i.e. the Spearman's Rank correlation coefficient was $< |0.7|$). Finally, we also included the mean temperature [29] and the distance from water (water bodies or water courses; [30]) due to their well-known influence on bat activity.

All variables included in the models were centered and scaled. The variables were uncorrelated (S1 Table in S1 File for the correlation matrix) and had variance inflation factor values < 3, showing no evidence for multicollinearity [31]. Co-variables were distributed homogeneously across the different datasets (S2 Table in S1 File). The distance variables (i.e. distance from water and distance from the nearest turbine) were log-transformed, as we expected the relationship between bat activity and distance variables to be logarithmic rather than linear [6,32,33]. We built a full model for each of the three datasets (low, moderate and high wind speed). Following multi-model inference [34], we generated from each full model a set of candidate models containing all possible variable combinations, yet limiting the number of variables to five to prevent overparameterization. We ranked candidate models by corrected Akaike information criterion (AICc) and retained the model with the lowest AICc, hereafter referred to as the best model. For each dataset, we ensured that the AICc of the best model was lower than the AICc of the null model. Full and best model compositions as well as results (estimates, standard errors and p-values) for the variables we tested are presented in SI Appendix, S3, S4 and S5 Tables in S1 File, respectively. We validated the best model results by checking their consistency with the results of all candidate models with ΔAICc <7 for each dataset. To do that, we checked that the candidate models mostly contained the same variables of interest (distance from wind turbine, wind incidence angle and their interaction) as the best model (S9 Fig in S1 File), and that the estimates, 95% confidence intervals, and p-values for these variables were not divergent from those of the best model variables (S10 Fig in S1 File).

We predicted bat activity based on the best models using the *predict* function (R package stats) on the centroid of 115,616 cells forming a grid around a fictional wind turbine, using a spatial resolution of 25 m (Fig 2). The resulting predictions range from 18 to 1499 meters for the distance from wind turbine, and 0 to 180˚ for the wind incidence angle.

Finally, when the interaction term (between the distance from the wind turbine and the wind incidence angle) was significant, we conducted post-hoc tests (emtrends, R package emmeans). We estimated the marginal means of linear trends of the effect of the distance from a wind turbine on bat activity, depending on whether the sampling site was located windward (0˚), lateral (90˚) or leeward (180˚) of the turbine.

## Results

We obtained 304,912 bat passes across the 154 sampled hedgerow sites: on average 845 (min: 6; max: 3804) bat passes per night for the low wind-speed dataset, 856 (min: 5; max: 4985) for the moderate wind-speed dataset, and 788 (min: 2; max: 4636) for the high wind-speed dataset. We found that wind speed and wind incidence angle are key elements to account for when assessing wind turbine effects on bats, as they strongly affected bat activity level and distribution. We present below the results from the best generalized linear mixed model for each of the three datasets separately (see the Materials and Methods section for more details about modeling approach).

### Bat response to wind turbine proximity in absence of wind

For the category of lowest wind speed, no or low wake effect was expected. Bat activity levels for a given distance from the turbine were in this case homogeneous around the turbine, as expected under low airflow disturbances. However, bat activity was significantly higher very near the turbine than further away, suggesting that bats were attracted to the wind turbine (Table 1).

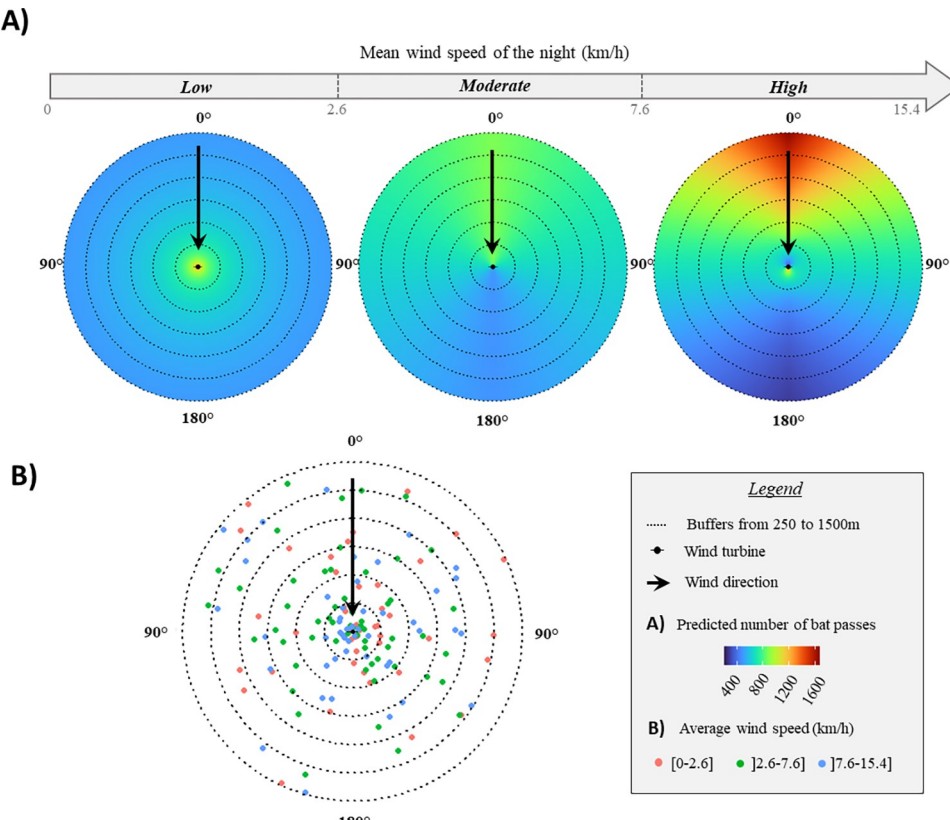

**Fig 2.** Predicted levels of activity of the bat *Pipistrellus pipistrellus* (A) and sampled points (B) around and up to 1500 m from the wind turbine for each wind-speed dataset per night. Predicted bat passes per night were 558 ± 66 (mean ± standard deviation) for the low wind-speed dataset ([0 to 2.6] km/h and up to 9.4 km/h considering wind gusts), 663 ± 116 for the moderate wind-speed dataset (]2.6 to 7.6] km/h and up to 18.6 km/h considering wind gusts) and 747 ± 308 for the high wind-speed dataset (]7.6 to 15.4] km/h and up to 30.4 km/h considering wind gusts).

## Bat response to wind turbine proximity and to the wake effect in windy conditions

In the presence of a wake effect in conditions of moderate wind speed (]2.6–7.6] km/h), bat activity was higher at small wind incidence angles (i.e. windward of the turbine) than at large angles. We no longer detected attraction as bat activity levels were similar near and far from the turbine (Table 1).

**Table 1. Predictors of bat activity near wind turbines: estimates ± standard errors and p-values (in italics) for the distance from the wind turbine, wind incidence angle and the interaction between them when retained in the best GLMM for each wind-speed dataset.** These results were consistent with most of the candidate models with ΔAICc <7 (S2 and S3 Figs in S1 File), confirming their robustness. NA = variable not retained in the best GLMM.

| | Low wind speed [0–2.6] km/h | Moderate wind speed ]2.6–7.6] km/h | High wind speed ]7.6–15.4] km/h |
|---|---|---|---|
| Log(Distance from wind turbine (m) + 1) | -0.208 ± 0.096 ***0.030*** | NA | -0.036 ± 0.107 *0.733* |
| Wind incidence angle (°) | NA | -0.177 ± 0.087 ***0.043*** | -0.199 ± 0.100 ***0.047*** |
| Log(Distance from wind turbine (m) +1) * Wind incidence angle (°) | NA | NA | -0.274 ± 0.100 ***0.006*** |

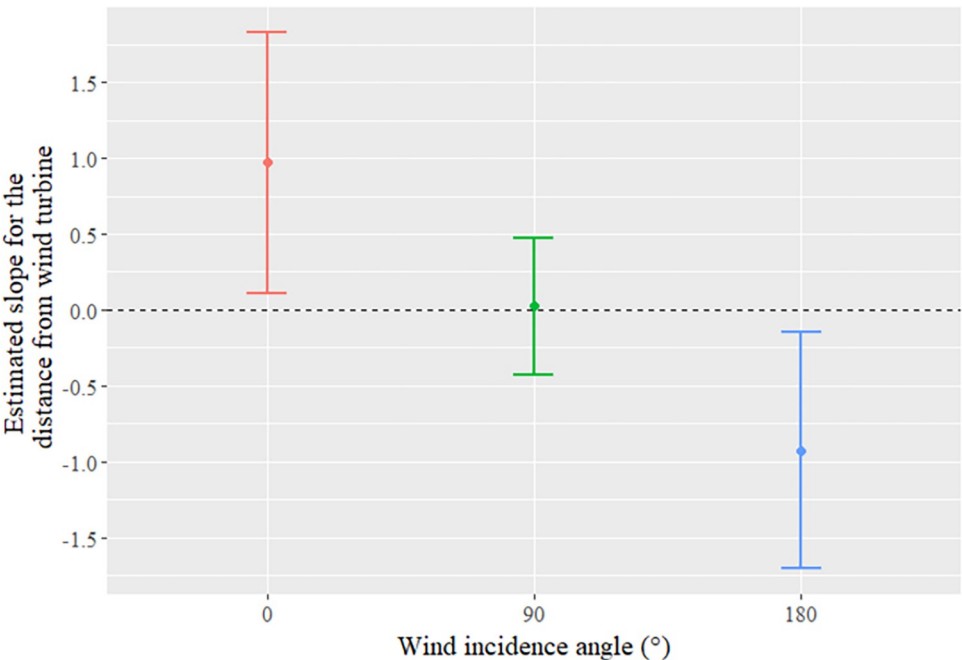

**Fig 3. Effect of the distance from the wind turbine on activity of the bat *Pipistrellus pipistrellus* windward (upwind; wind incidence angle 0˚), leeward (downwind; 180˚) and lateral (to the side; 90˚) of the turbine.** Post-hoc tests were conducted on the significant interaction between the distance from the wind turbine and the wind incidence angle for the high wind-speed dataset (]7.6–15.4] km/h) to estimate marginal means of linear trends (emtrends, R package emmeans) of the effect of distance to wind turbine for the different wind incidence angles. The error bars correspond to a confidence interval of 95%.

In conditions of high wind speed (]7.6–15.4] km/h), the distribution of bat activity was strongly structured by airflow turbulence. Globally, we observed higher activity at small wind incidence angles (i.e. windward the turbine) than at large angles (i.e. leeward the turbine). Bat activity was also affected by distance from the wind turbine, in different ways depending on the wind incidence angle (interaction term, Table 1). Leeward of the turbine, activity was higher near the turbine, suggesting that bats were attracted to the turbine where turbulence was low and, at rotor height, wind speed was minimal [35]. In contrast, on the windward side of the turbine, bat activity was higher further away from the turbine than close to it (avoidance, shown along the distance gradient). A complementary analysis conducted on leeward and windward subsets of the high wind-speed dataset confirmed that both attraction leeward of the turbine and avoidance windward of the turbine were significant (S6 Table in S1 File). Finally, bat activity was homogeneous along the gradient of distance from the wind turbine (no behavioral response) lateral of the turbine in relation to wind direction (wind incidence angle of 90˚; Fig 3). The interaction between distance from the turbine and wind incidence angle reveal the simultaneous coexistence of attraction and avoidance responses, depending entirely on the airflow's spatial structure. Bats windward of the turbine avoided it along the longitudinal distance gradient, while bats also avoided the turbulent leeward wake area on the circular azimuth gradient.

## Discussion

Our results confirm that bats are more disturbed downwind the turbine that upwind as evidenced by low bat activity in the wake area. Thus, they are consistent both with the hypothesis

that airflow disturbances generated by spinning wind turbine blades strongly affected activity distribution of *P. pipistrellus* around turbines, and with the alternative hypothesis that the noise produced by the operating turbines could deter bats, as it has been shown that some noise can affect bat activity [36–38]. However, whether the pattern of propagation of turbine-generated noise is the same as the one of the wake distribution patterns remains to be tested and the hypothesis that this noise could affect bats in a way causing either attraction or avoidance seems rather unlikely as the few studies conducted on the topic have shown that potential ultrasonic sounds emitted by a turbine attenuate on short distances [39]. Our results thus reveal that the conflicting effects of wind turbines on bats so far reported in the literature–both attraction and avoidance–may be explained by variation in airflow disturbances, so that accounting for these disturbances is essential when studying and mitigating effects of wind turbines on bats. Furthermore, the fact that we detected attraction all around the turbine only in conditions of no wind or low wind speed and not for moderate or high wind speed, while avoidance was detected only at high wind speeds (i.e. not at low or moderate wind speed), provides some clues about the nature of the underlying mechanisms. Our findings should considerably improve predictions regarding the area of major risks of habitat loss for bats (i.e. downwind the turbine) and thus mitigation of the impacts of wind turbines on bats by taking into account the location of the wind turbine regarding both prevailing wind and bat habitat in wind energy planning. These results could also be considered in deterrent technology placement for relatively high wind speed, although this measure efficiency for *P. pipistrellus* remains to be confirmed [40].

In the absence of wind and wind-related turbulence around the turbine, *P. pipistrellus* activity levels were homogeneous at all wind incidence angles. However, when the wind was blowing and the blades spinning, *P. pipistrellus* activity was relatively higher windward of the turbines, suggesting, as hypothesized, that bats avoid disturbed airflow wake areas. Indeed, bats probably struggle even more to fly in turbulence generated by wind turbines than in undisturbed conditions on the windward side of linear elements. Hence, bats concentrate in areas less difficult to fly through, namely areas where the wind speed is lower or the turbulence less strong.

*P. pipistrellus* may avoid turbulent wake areas where flight conditions are suboptimal because of increased flight costs there, when commuting or foraging [41]. Alternatively, they may perceive these areas as an unsafe part of the 'landscape of fear', defined as "the way the animal perceives its environment based on the cost-benefit analysis of the trade-off of food and safety" [42]. Bats may indeed perceive turbulent wake areas as more exposed to predation risk, as their flight is likely to be more affected by the wind than the flight of predators that are often bigger and faster than them [43]. For instance, *P. pipistrellus*, whose speed range is approximately of 11.5–20.9 km/h [43], would achieve only 40% to 60% of its still-air range against a headwind of 9 km/h. A third reason for bats to avoid areas of turbulence is that prey availability may be lower in these areas. Indeed, flying insects are likely to seek calmer areas, as they are known to concentrate leeward of windbreaks [44].

At high wind speeds (on average 7.6 to 15.4 km/h per night in our dataset), we detected more *P. pipistrellus* activity far away from the wake area than close to it, suggesting an avoidance of the wake area. However, this effect interacted significantly with the effect of distance from the turbine. In these very windy conditions, *P. pipistrellus* activity windward of the turbine was higher far away from it than close to it (avoidance of the turbine). The fact that avoidance of the turbine was only detected at high wind speeds suggests that this response is related to fast wind turbine operation rather than to wind turbine structure. In such disturbed conditions, bats could perceive these turbulent areas at landscape scales as suggested by Cryan et al. 2014, explaining this avoidance of the turbine windward the turbine. It is consistent with a

previous study showing avoidance of small wind turbines only when blades were spinning [45]. Both avoidance along the distance gradient, which occurred only windward of the turbine, and avoidance of the turbulent wake area leeward of the turbine could result from the avoidance of turbulent areas, since some turbulence is generated by spinning blades on the leeward side. Bats can sense airflows and orient themselves by using airflows, thanks to aerodynamic feedback from tactile receptors associated with their wing hairs [14]. At the landscape scale, bats may use airflows to orient themselves in relation to wind turbines [15], but airflow paths may be disturbed or may no longer exist in the turbulent leeward wake of the turbine. Alternatively, bats may avoid the wake area simply due to their perception of suboptimal flight conditions there, as discussed above.

Close to the turbine, we detected relatively high *P. pipistrellus* activity in two situations: at low wind speeds, regardless of the wind incidence angle; and at high wind speeds, only leeward of the turbine. This attraction under different wind conditions suggests that the mechanisms involved are different. At low wind speeds, this response is likely to be linked to the structure of the wind turbine itself rather than to its operation [46], which is consistent with the hypothesis that bats are attracted to turbines due to confusion with tall trees or while seeking foraging and drinking opportunities at the turbine mast [39]. The concentration of bat activity just leeward of the turbine at high wind speeds is consistent with observations that most bats approach the nacelle from the leeward side at the local scale ([15]; evidenced by thermal cameras at 12 m from the turbine). However, this pattern of approach was mostly found for stationary or very slow spinning blades [15], while the blades in our dataset were mostly spinning. Therefore, our observations cannot be explained by the idea that bats perceive turbines with stationary or slow spinning blades as trees due to similar airflow profiles around the structures [15]. At high wind speeds when blades are spinning fast, we suggest that bats may use the area just leeward of the nacelle and possibly the mast to shelter from the wind as they seem to do with the leeward side of tree lines [47], given that wind speed is greatly reduced there. It is also possible that flying insects shelter in this area, making it an interesting foraging area for bats. Additionally, bats flying near the ground in this area (i.e. leeward and close to the turbine) may benefit from not being affected by the turbulence which is rather concentrated at the rotor height at this distance from the mast [12,48]. However, we have no information on the altitude of bats concentrated just leeward of the turbine, an aspect that seems important to investigate in future studies to confirm or refute the increased exposure to collisions with blades due to this attraction.

The avoidance of wind turbines by bats is determined at least partially by the habitat [5], which may suggest a constant response for a given habitat. However, here we reveal greater complexity: responses of bats can vary within the same habitat (the hedgerow), depending on abiotic conditions and specifically airflows. This has considerable implications for wind energy planning, suggesting that plans for wind farms should be adapted to both the type of habitat and to the wind farm's location in relation to prevailing wind directions (i.e. the likely exposure time of this habitat to the wake effect). In particular, we strongly recommend avoiding configurations involving the installation of a turbine between the origin of prevailing winds and important habitats for bats, such as hedgerows, water or woodlands. The area leeward of an operating wind farm is the most affected by the loss of habitat, probably particularly in the near-wake area (to a distance of 2 to 4 times the rotor diameter [12]) and bats may also be attracted to the turbine in this area, which is likely to increase fatality risks.

Further studies should assess whether these findings can be generalized to all bat species or not. We expect airflow disturbances to affect slower flying animals more [43], which are also, in many cases, the smaller ones [43]. It would be relevant to conduct a similar study in autumn and in the end of the summer, when peak of bat activity near the turbine [49] and peak of

fatalities [50] occur, to determine whether bats respond to wind turbine wakes in the same way (i) in a migration context and (ii) on windier nights (more windy hours). Indeed, bat behavior and habitat use can be different in these periods, compared to the reproduction period, involving for example movements along one prevailing direction. Thus, bat sensitivity to the wake effect could vary. It could be also useful to assess bat response to wake effect at different distances to hedgerows, as we would expect bat to fly closer to hedgerows in windy and/or disturbed airflow conditions [47]. Finally, in this study we used average wind speed and prevailing wind direction as a proxy for the location and intensity of the wake effect and we measured bat activity at ground. Future studies should be conducted at a finer scale by including metrics of wind turbulence, considering the wake effect in 3D, simultaneously measuring bat activity at both ground and at height and maybe even pairing acoustic recorders with thermal cameras. This would allow to better understand how the wake effect affects bat distribution in two dimensions–horizontally and vertically. A finer temporal scale could also be considered to test whether bats adapt their foraging or commuting habitat through the night depending on the wind direction and whether the impact of the wake is stronger in hours of high activity.

Given the global context of wind energy development [50] and the latest report of the Intergovernmental Panel on Climate Change, which presents wind energy as one of the most efficient ways to reduce greenhouse gas emissions [51], levers to understand and mitigate the impacts of wind turbines on biodiversity are more urgent than ever. This is especially true for airborne vertebrates, populations of which can be threatened by wind energy installations [52,53]. Based on the evidence we present here, we therefore encourage the consideration of abiotic conditions, especially airflows, in wind energy planning and mitigation strategies.

## Supporting information

**S1 File. SI Acoustic detection range. SI Statistical analysis—Covariable extraction. SI Statistical analysis—Model weight calculation. S1 Fig**. Map showing all 776 wind turbines in the studied counties (within Bretagne and Pays-de-la-Loire regions, western France), the land cover of the area, and the 154 sampled sites. **S2 Fig**. Boxplot showing that a large gradient of distance from the wind turbine was sampled each night. Horizontal line: median; box: first and third quartiles; whiskers: range; dots: outliers. **S3 Fig**. Boxplot of the wind incidence angles sampled each night. Various wind incidence angles were sampled each night to obtain a gradient of location around the turbine in relation to wind direction. Horizontal line: median; box: first and third quartiles; whiskers: range; dots: outliers. **S4 Fig**. Distribution of average wind turbine blade speed rotation per night for the entire dataset (left) and for the three subdatasets (right). **S5 Fig**. Distribution of the interaction of the tested gradients (wind incidence angle, depending on the distance from the wind turbine) for the entire dataset (left) and for the three subdatasets (right). **S6 Fig**. Distribution of average wind gusts per night for the entire dataset (left) and for the three subdatasets (right). **S7 Fig**. Distribution of average wind speed per night for the entire dataset (left) and for the three subdatasets (right). **S8 Fig**. Number of *Pipistrellus pipistrellus* each night depending on the average wind speed of the night (i.e. for the three subdatasets). **S9 Fig**. Candidate models within a ΔAICc < 7 containing (in color) or not (in grey) the variables of interest: distance to wind turbine (blue, on the left), wind incidence angle (orange, in the center), and their interaction (green, on the right). **S10 Fig**. Estimates, 95% confidence intervals, and p-values for the variables of interest contained in each candidate model within a ΔAICc < 7. **S1 Table.** Correlation matrix between variables included in the models for all datasets. No variables were correlated (r < |0.7|). Besides this correlation check, we checked for potential collinearity problems in the full models using the Variance Inflation Factor (VIF) before modelling (R package performance). WT = wind turbine. **S2 Table.**

Mean ± standard deviation (min.-max.) of all variables included in the statistical analysis. **S3 Table.** AICc and $R^2$ of null, full and best models for each dataset. **S4 Table.** Estimates ± standard errors and p-values for the variables of interest in the full and best models (GLMMs). **S5 Table.** Estimates ± standard errors and p-values for the co variables in the full and best models (GLMMs). **S6 Table.** Estimates ± standard errors and p-values (in italics) for the predictors of bat activity for the model resulting from a complementary analysis. (DOCX)

## Acknowledgments

We thank IN2P3 Computing Centre and PCIA-MNHN for providing computing and storage facilities to process and archive in the long term all the bat recordings used in this study; Yves and Didier Bas for their help with the archiving; wind farm developers, operators and owners, for providing us with information on features of the wind turbines and blade rotation speeds for the nights on which we sampled; Nancy Jennings for the editing service; and Jérémy Prouff, Anne-Laure Brissard and Jean-Noël Caron for helping with the figures.

## Author Contributions

**Conceptualization:** Camille Leroux, Kévin Barré, Christian Kerbiriou, Isabelle Le Viol.

**Data curation:** Camille Leroux.

**Formal analysis:** Camille Leroux, Kévin Barré, Christian Kerbiriou, Isabelle Le Viol.

**Funding acquisition:** Nicolas Valet, Christian Kerbiriou, Isabelle Le Viol.

**Investigation:** Camille Leroux, Kévin Barré, Christian Kerbiriou, Isabelle Le Viol.

**Methodology:** Camille Leroux, Kévin Barré, Christian Kerbiriou, Isabelle Le Viol.

**Project administration:** Nicolas Valet, Christian Kerbiriou, Isabelle Le Viol.

**Resources:** Nicolas Valet.

**Supervision:** Kévin Barré, Nicolas Valet, Christian Kerbiriou, Isabelle Le Viol.

**Validation:** Camille Leroux, Kévin Barré, Christian Kerbiriou, Isabelle Le Viol.

**Visualization:** Camille Leroux, Kévin Barré, Christian Kerbiriou, Isabelle Le Viol.

**Writing – original draft:** Camille Leroux, Kévin Barré, Christian Kerbiriou, Isabelle Le Viol.

**Writing – review & editing:** Camille Leroux, Kévin Barré, Nicolas Valet, Christian Kerbiriou, Isabelle Le Viol.

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
