## [Decision Letter · Decision Letter 0]

24 Nov 2023

PONE-D-23-31327Distribution of common pipistrelle (* Pipistrellus pipistrellus *) activity is altered by airflow disruption generated by wind turbinesPLOS ONE

Dear Dr. Leroux,

Thank you for submitting your manuscript to PLOS ONE. After careful consideration, we feel that it has merit but does not fully meet PLOS ONE’s publication criteria as it currently stands. Therefore, we invite you to submit a revised version of the manuscript that addresses the points raised during the review process.

We look forward to receiving your revised manuscript.

Kind regards,

A A Chowdhury, Ph.D., FHEA

Academic Editor

PLOS ONE

“We thank the Association Nationale de la Recherche et de la Technologie (Grant No. 2019/1566) and Auddicé biodiversité for funding this research; IN2P3 Computing Centre and PCIA-MNHN for providing computing and storage facilities to process and archive in the long term all the bat recordings used in this study; Yves and Didier Bas for their help with the archiving; wind farm developers for funding part of the bat recorders; wind farm developers, operators and owners, for providing us with information on features of the wind turbines and blade rotation speeds for the nights on which we sampled; Nancy Jennings for the editing service; and Jérémy Prouff, Anne-Laure Brissard and Jean-Noël Caron for helping with the figures. Kévin Barré was funded by the Agence de la transition écologique (ADEME), Christian Kerbiriou by Sorbonne University, and Isabelle Le Viol by the french National Museum of Natural History (MNHN).”

“We thank the Association Nationale de la Recherche et de la Technologie (Grant No. 2019/1566) and Auddicé biodiversité for funding this research; IN2P3 Computing Centre and PCIA-MNHN for providing computing and storage facilities to process and archive in the long term all the bat recordings used in this study; Yves and Didier Bas for their help with the archiving; wind farm developers for funding part of the bat recorders; wind farm developers, operators and owners, for providing us with information on features of the wind turbines and blade rotation speeds for the nights on which we sampled; Nancy Jennings for the editing service; and Jérémy Prouff, Anne-Laure Brissard and Jean-Noël Caron for helping with the figures. Kévin Barré was funded by the Agence de la transition écologique (ADEME), Christian Kerbiriou by Sorbonne University, and Isabelle Le Viol by the french National Museum of Natural History (MNHN).”

“We thank the Association Nationale de la Recherche et de la Technologie (Grant No. 2019/1566) and Auddicé biodiversité for funding this research; IN2P3 Computing Centre and PCIA-MNHN for providing computing and storage facilities to process and archive in the long term all the bat recordings used in this study; Yves and Didier Bas for their help with the archiving; wind farm developers for funding part of the bat recorders; wind farm developers, operators and owners, for providing us with information on features of the wind turbines and blade rotation speeds for the nights on which we sampled; Nancy Jennings for the editing service; and Jérémy Prouff, Anne-Laure Brissard and Jean-Noël Caron for helping with the figures. Kévin Barré was funded by the Agence de la transition écologique (ADEME), Christian Kerbiriou by Sorbonne University, and Isabelle Le Viol by the french National Museum of Natural History (MNHN).”

4. We note that Figure 1 in your submission contain copyrighted images. All PLOS content is published under the Creative Commons Attribution License (CC BY 4.0), which means that the manuscript, images, and Supporting Information files will be freely available online, and any third party is permitted to access, download, copy, distribute, and use these materials in any way, even commercially, with proper attribution. For more information, see our copyright guidelines: http://journals.plos.org/plosone/s/licenses-and-copyright.

Reviewers' comments:

**Comments to the Author**

1. Is the manuscript technically sound, and do the data support the conclusions?

Reviewer #1: Yes

Reviewer #2: Yes

Reviewer #3: Yes

2. Has the statistical analysis been performed appropriately and rigorously? 

Reviewer #1: Yes

Reviewer #2: N/A

Reviewer #3: Yes

3. Have the authors made all data underlying the findings in their manuscript fully available?

Reviewer #1: Yes

Reviewer #2: Yes

Reviewer #3: Yes

4. Is the manuscript presented in an intelligible fashion and written in standard English?

Reviewer #1: Yes

Reviewer #2: Yes

Reviewer #3: Yes

5. Review Comments to the Author

Reviewer #1: 1) Suggest removing barotrauma. There are several studies showing barotrauma to be an unlikely cause of mortality in bats. See Lawson et al. 2020, Rollins et al. 2012, and Grodsky et al. 2011.

2) The description of the results and following discussion are rather confusing. I think a figure similar to figure 2, that summarize the actual bat activity would go a long way in simplifying the results.

3) The first couple of sentences in the discussion create doubt in the authors assessment of whether their findings are a result of wind speed patterns or sound. Since they did not account for sound in their study, it cannot be ruled out as a potential factor for attraction/avoidance.

4) Line 249: '...avoidance only at high wind speeds provides clues about the nature of the underlying mechanisms.' However, in lines 216-218 the authors state that bat activity was higher near the turbine under high wind speed conditions. So, its not simply avoidance at high wind speeds.

5) Line 250-251: 'Our findings should considerably improve predictions and thus mitigation of the impacts of wind turbbine on bats.' How?

6) Line 270-271:'At high wind speeds..., we detected more P. pipistrellus activity far away from the wake area than close to it.' This statement contradicts lines 216-218.

7) Starting at line 293 through 297 there is reference to (24). Assuming this is intended to be a reference, perhaps Cryan et al. 2014?

8) Apologies if I missed this, but I dont recall seeing an explanation for why bats would avoid wind turbines from the windward side under high wind speed conditions.

Reviewer #2: The study relies on a robust dataset from bat passes around the wind turbines in western France. They tested new approaches to how wind turbines affect bat activity, installing autonomous recorders in different places around the turbines to see if the bats are avoiding or are attracted by the turbines, which is innovative and exciting. Although I am a little worried about some statistical analyses used and I highly recommend clarifying some aspects. The introduction is consistent and well-written. The discussion addresses the results found in the study, and it seems that the authors did a good job in the literature review. I miss more figures of the results, and I recommend transforming the contents of the table into graphs to better visualization.

I have specific concerns during the text, and you can see below:

L88: Fig. 1. (B) The legends are tiny.

L121: I do not know this software and I don’t know how this software is efficient in identifying automatically the bat species. So I recommend citing other published studies that used this software for the automatic identification of the data.

L121-123: As I am a bat ecologist not european, I suggest talking more about the bat species that you chose. Which is the foraging strategy of this species? Diet and height of the foraging?

L149: Did you include the sampling nights as a random effect on the models? I saw in the Supplementary Material that you included (1|site) + (1|night) in the full models (Table S4). Why do you include these two variables separated from each other? Did you test if these random effects are the best for the models?

L150: I miss the specifications of the family distribution chosen by each model. Did you use “DHARMa” package to check the model residuals as well as the overdispersion and zero inflation? It is common that these datasets of bat activity are overdispersed.

L191-228: I suggest clarifying why you analyzed the AIC models with all variables but included only the GLMM models the distance from the wind turbine, wind incidence angle, and interaction. For example, I saw in Table S4 that mean temperature was a variable that was presented in the dataset of all wind speeds, low and high speed. If these variables were presented in the best models of AIC, it is possible to explain a little bit of the effects?

Table 1. I recommend transforming the results of the table into a graph/figure. May the coefficient plots or Dot and Whisker plots be formats for these results?

L330: Another aspect that is important to mention is the hourly variation of bat activity. The effect of high-speed turbines may be stronger at the beginning of the night when the bat activity is higher. I recommend discussing how this finer temporal scale can affect your results. (May you can test this in a future study?!)

Reviewer #3: General Comments:

The authors test if the wake effect changed Pipistrellus pipistrellus bat acoustic activity patterns around operating wind turbines in western France in June using acoustic detectors at varying distances from wind turbines. They predicted less bat activity on the leeward side of the wind turbine during wind speeds when wake effect would be moderate to large. They do a thorough statistical analysis of the data and provide adequate discussion of results. This paper contributes to our understanding of how the wake effect could play a role in where bat activity occurs during high-risk periods around wind turbines. It also adds more evidence for attraction to wind turbines. I recommend this manuscript for publication with some improvements, particularly to the discussion.

Discussion:

In lines 243 to 245 it is stated that turbine-generated noise affecting bats has not been tested, but this is not accurate. There are several studies conducted in North America (refer to Guest et al. 2022 and the section on noise) that could be discussed and cited. If the authors mean to say that turbine-generated noise has not been test for this species or in Europe, then the text should be altered to specify.

The discussion lacks any implications on deterrent technology placement, which could strengthen the conclusions. I recommend expanding on this with the findings that during the highest risk periods (when wind turbines are operational at moderate to high wind speeds) bats are most acoustically active on the windward side. This could inform on deterrent placement particularly in areas with a dominant prevailing wind direction.

For future recommendations to further this type of research, is there any reason to believe this pattern would be different in August when peak fatality occurs? I think it would be helpful to include comment here. Additionally, pairing acoustics with cameras would allow for some assessment of abundance and reduce limitations of acoustic analyses (e.g., is it a single bat versus 100; some bats don’t always echolocate and go undetected acoustically).

General Comments/Edits:

Many brackets are in the wrong location or are the incorrect facing brackets throughout results and tables. Font changes throughout and citation style is not consistent. Suggest a thorough editorial review.

Lines 192-195 are restating the methods and can be removed.

Lines 195-198-you can’t have negative bat passes so it might be better to state the range instead of the SD here.

Line 211-remove colon and replace with “and”

Lines 293-297 there is a switch in citation style to numbers.

Lines 318-319 seems like a good place for a new paragraph.

6. PLOS authors have the option to publish the peer review history of their article (what does this mean?). If published, this will include your full peer review and any attached files.

Reviewer #1: No

Reviewer #2: No

Reviewer #3: No

---

## [Author Response · Author response to Decision Letter 0]

7 Feb 2024

Response to Reviewers and editor comments are uploaded in the "Response to Reviewers" document.

---

## [Decision Letter · Decision Letter 1]

12 Apr 2024

Distribution of common pipistrelle (* Pipistrellus pipistrellus *) activity is altered by airflow disruption generated by wind turbines

PONE-D-23-31327R1

Dear Dr. Leroux,

We’re pleased to inform you that your manuscript has been judged scientifically suitable for publication and will be formally accepted for publication once it meets all outstanding technical requirements.

Kind regards,

Ashfaque Ahmed Chowdhury, Ph.D., FHEA, FIEB

Academic Editor

PLOS ONE

Reviewers' comments:

Reviewer's Responses to Questions

**Comments to the Author**

1. If the authors have adequately addressed your comments raised in a previous round of review and you feel that this manuscript is now acceptable for publication, you may indicate that here to bypass the “Comments to the Author” section, enter your conflict of interest statement in the “Confidential to Editor” section, and submit your "Accept" recommendation.

Reviewer #1: (No Response)

Reviewer #3: All comments have been addressed

2. Is the manuscript technically sound, and do the data support the conclusions?

Reviewer #1: Partly

Reviewer #3: Yes

3. Has the statistical analysis been performed appropriately and rigorously? 

Reviewer #1: Yes

Reviewer #3: Yes

4. Have the authors made all data underlying the findings in their manuscript fully available?

Reviewer #1: Yes

Reviewer #3: Yes

5. Is the manuscript presented in an intelligible fashion and written in standard English?

Reviewer #1: Yes

Reviewer #3: (No Response)

6. Review Comments to the Author

Reviewer #1: (No Response)

Reviewer #3: The authors addressed my comments sufficiently and I recommend for publication. The work is relevant and important for the field.

7. PLOS authors have the option to publish the peer review history of their article (what does this mean?). If published, this will include your full peer review and any attached files.

Reviewer #1: No

Reviewer #3: No

---

## [Editor Report · Acceptance letter]

29 Apr 2024

PONE-D-23-31327R1 

PLOS ONE

Dear Dr. Leroux, 

I'm pleased to inform you that your manuscript has been deemed suitable for publication in PLOS ONE. Congratulations! Your manuscript is now being handed over to our production team.

Kind regards, 

on behalf of

Dr. Ashfaque Ahmed Chowdhury 

Academic Editor

PLOS ONE